# Environmental Factors Associated with Physical Activity in Rural U.S. Counties

**DOI:** 10.3390/ijerph18147688

**Published:** 2021-07-20

**Authors:** Christiaan G. Abildso, Shay M. Daily, M. Renée Umstattd Meyer, Michael B. Edwards, Lauren Jacobs, Megan McClendon, Cynthia K. Perry, James N. Roemmich

**Affiliations:** 1Department of Social and Behavioral Sciences, School of Public Health, West Virginia University, Morgantown, WV 26506, USA; 2WVU Office of Health Affairs, Robert C. Byrd Health Sciences Center, West Virginia University, Morgantown, WV 26505, USA; smdaily@hsc.wvu.edu; 3Department of Public Health, Robbins College of Health and Human Sciences, Baylor University, Waco, TX 76798, USA; Renee_Umstattd@baylor.edu (M.R.U.M.); megan.mcclendon@ag.tamu.edu (M.M.); 4Department of Parks, Recreation and Tourism Management, College of Natural Resources, North Carolina State University, Raleigh, NC 27695, USA; mbedwards@ncsu.edu; 5School of Kinesiology and Physical Education, College of Education and Human Development, University of Maine, Orono, ME 04469, USA; lauren.jacobs@maine.edu; 6School of Nursing, Oregon Health Science University, Portland, OR 97239, USA; perryci@ohsu.edu; 7US Department of Agriculture, Agricultural Research Service, Grand Forks, ND 58201, USA; james.roemmich@usda.gov

**Keywords:** rural health, physical activity, positive deviance, social-ecological

## Abstract

Background: Rural U.S. adults’ prevalence of meeting physical activity (PA) guidelines is lower than urban adults, yet rural-urban differences in environmental influences of adults’ PA are largely unknown. The study’s objective was to identify rural-urban variations in environmental factors associated with the prevalence of adults meeting PA guidelines. Methods: County-level data for non-frontier counties (*n* = 2697) were used. A five-category rurality variable was created using the percentage of a county’s population living in a rural area. Factor scores from Factor Analyses (FA) were used in subsequent Multiple Linear Regression (MLR) analyses stratified by rurality to identify associations between environmental factor scores and the prevalence of males and females meeting PA guidelines. Results: FA revealed a 13-variable, four-factor structure of natural, social, recreation, and transportation environments. MLR revealed that natural, social, and recreation environments were associated with PA for males and females, with variation by sex for social environment. The natural environment was associated with PA in all but urban counties; the recreation environment was associated with PA in the urban counties and the two most rural counties. Conclusions: Variations across the rural-urban continuum in environmental factors associated with adults’ PA, highlight the uniqueness of rural PA and the need to further study what succeeds in creating active rural places.

## 1. Introduction

Rural U.S. communities experience a multitude of impactful health disparities [1], including a lower prevalence of engaging in preventive health behaviors such as physical activity (PA) [2]. Specifically, the percentage of adults meeting combined aerobic and muscle-strengthening PA guidelines is 31% lower among rural residents compared to urban residents [3]. This deficit exists despite evidence of greater relative increases of meeting PA guidelines from 2008 to 2017 among rural adults (47.4%) compared to urban adults (30.4%) overall. Analyses by Whitfield and colleagues also revealed greater relative increases in rural adults across multiple sociodemographic categories, including males and females, all age groups except 25–34 years, all Census regions except the South, those with a high school or some college educational attainment, and in Black and White non-Hispanic adults [3]. While these relative improvements are promising, the prevalence of meeting the PA guidelines is still greater among urban adults than rural adults of all races and ethnicities, for both sexes, and across all regions and education levels.

Two decades of PA research has accumulated evidence that multiple “environments” influence PA based on a social-ecological perspective [4], emphasizing the primacy of the physical, or built, environment [5] in the multiple places that individuals interact with daily (e.g., work, school, home), and the transportation system that connects those places. As illustrated by Sallis and colleagues [6] built environments are influenced by political, natural, and social/cultural environments. Collectively, these environments influence—and are influenced by—individual decisions in a cycle of reciprocal determinism [7] or mutual influence [4]. Much of what is known about the environmental influences on PA is based on studies of urban and suburban built environments that may have little applicability to rural places. Recent literature reviews consistently highlight this gap, identifying the near-exclusive reliance on urban-derived evidence to inform practices in rural communities and making strong recommendations for research specific to rural communities, given the health and PA disparities observed in rural communities. Despite these recommendations, there has yet to be an empirical analysis of objective measures on a national level examining the rural-urban variation of environmental associations with PA [8,9,10]. 

Rural-urban comparisons are important so that appropriate interventions can be developed and undertaken. Recent research highlights some of the rural-urban differences in walking behaviors and perceived walkability [3,11,12]. For example, Whitfield and colleagues [11] found that walking is associated with the number of environmental supports (e.g., sidewalks) and destinations in both urban and rural areas of the U.S., but the most influential types of supports and destinations may differ between urban and rural communities. Additionally, even if the associations between walkability supports and behaviors are similar across rural and urban communities, the implementation of supports for walking may face unique challenges in rural communities (e.g., longer block lengths, lower development density, faster-moving traffic, the availability of ample parking) [13]. While access to natural amenities that facilitate PA like walking or biking (e.g., multiuse trails) may be associated with higher PA in rural communities [14,15], it is also possible that natural environments such as weather and climate may present meaningful barriers to PA among rural residents in comparison to urban residents [16]. Research further suggests that sociocultural factors (e.g., crime, safety, social support) may be associated with PA prevalence in rural communities, though few studies have compared the relationship between these factors and PA across rural and urban settings [10,17]. A childhood obesogenic environment index (COEI) inclusive of social-ecological variables hypothesized to be associated with sedentary, nutrition, and PA behaviors suggests that rural areas are more conducive to obesogenic behaviors of children than urban and micropolitan counties in all regions of the U.S. [18,19]. However, an analysis of national-level data to identify the social, natural, built, and economic environment factors associated with rural adults’ PA in the U.S. has yet to be undertaken.

The objective of this study was to answer the following question: how do environmental factors associated with the prevalence of adults meeting PA guidelines differ along the rural-urban continuum for males and females? 

## 2. Materials and Methods 

### 2.1. Procedures 

Data from 3140 counties and county-equivalents (e.g., Baltimore City) (hereafter “counties”) were utilized for analyses. Rurality was defined as the county population living in rural areas according to 2010 Census data. Frontier counties (*n* = 443), defined as counties with a population density of six or fewer people per square mile [20], were excluded from analyses. The 2697 counties in the analytic sample were classified into five groups. The most rural group (“Exclusively Rural”) was defined as counties in which 100% of the population was living in rural areas. The remaining 2308 counties were then divided equally into quartiles based on the percentage of the population living in rural areas. The final sample resulted in 2697 observations from all 50 U.S. States, categorized as *Exclusively Rural* (*n* = 389), *Rural* (*n* = 577), *Somewhat Rural* (*n* = 577), *Somewhat Urban* (*n* = 577), and *Urban* (*n* = 577).

### 2.2. Data Sources

This study included a series of indicators from eight publicly available national data sources. For brevity, only a brief description for each data source has been provided, with URLs where additional methodologies, questionnaires, metrics, and data information may be found (see Table 1). Data collection years ranged from 2010 to 2011 to match the timeframe of the collection of the primary dependent variable (see Section 2.3.1).

#### 2.2.1. The American Community Survey (ACS)

The ACS is a critical element in the Census Bureau’s reengineered decennial census program. The ACS collects and produces population and housing information every year and publishes both one-year and five-year estimates [21].

#### 2.2.2. The Behavioral Risk Factor Surveillance (BRFSS)

The BRFSS is a state-based surveillance system that is operated by state health departments in collaboration with the Centers for Disease Control and Prevention (CDC). The BRFSS collects self-reported information by telephone on an assortment of behaviors and health outcomes among adults [22].

#### 2.2.3. The FBI Uniform Crime Reporting (UCR) Program

The UCR Program consists of four data collections from roughly 17,000 law enforcement agencies in the U.S.: (1) the National Incident-Based Reporting System (NIBRS), (2) the Summary Reporting System (SRS), (3) the Law Enforcement Officers Killed and Assaulted (LEOKA) Program, and (4) the Hate Crime Statistics Program. The UCR Program publishes annual reports for each of these data collections, a preliminary semiannual report of summary data each winter, and statistical publications, such as the County-Level Detailed Arrest and Offense Data report that is utilized to create the County Health Rankings [23].

#### 2.2.4. Fatality Analysis Reporting System (FARS)

The FARS is a census of fatal motor vehicle crashes with a set of data files documenting all qualifying fatalities that occurred within the 50 States, the District of Columbia, and Puerto Rico since 1975 [24].

#### 2.2.5. National Environmental Public Health Tracking Network (EPH)

For more than a decade, the EPH Program has collected, integrated, and analyzed non-infectious disease and environmental data from a nationwide network of partners. The purpose of this program is to deliver information and data to protect the nation from health issues arising from or directly related to environmental factors [25].

#### 2.2.6. 2010. Decennial United States Census

The decennial U.S. Census gathers data from all households in the United States and Island Areas related to name, gender, age, race, ethnicity, relationship, and home-ownership. The 2010 questionnaire was one of the shortest in history—asking just 10 questions—and is augmented by the U.S. Census Bureau’s annual ACS [21].

#### 2.2.7. CDC Wide-Ranging Online Data for Epidemiologic Research System (CDC WONDER)

CDC WONDER manages nearly 20 collections of public-use data for U.S. births, deaths, cancer diagnoses, tuberculosis cases, vaccinations, environmental exposures, and population estimates, among other topics. These data collections are available as online databases, which provide public access to ad-hoc queries, summary statistics, maps, charts, and data extracts. Most of the data are updated annually; some collections are updated monthly or weekly [26]. 

### 2.3. Measures

Table 1 provides a description, data source and collection year, and definition of each county-level variable and PA prevalence. 

#### 2.3.1. Prevalence of Physical Activity

For the current study, the county-level prevalence of meeting the PA guidelines by males and females over 18 years old, assessed via the 2011 BRFSS, was the outcome of interest. Individual respondents were asked about the frequency and time of each type of PA that they engaged in outside of work in the prior month if they engaged in any activity. The relative intensity of each type of activity engaged in was classified as moderate or vigorous based on the age and sex of the respondent. Each respondent was classified as meeting PA guidelines, insufficiently active, or inactive based on their responses. At the time of data collection for the 2011 BRFSS, the U.S. PA guidelines were for adults to achieve 150 min of moderate intensity PA per week, the equivalent in vigorous PA, or a combination of moderate and vigorous PA where one minute of vigorous PA is equivalent to two minutes of moderate PA. At the county level, the prevalence estimates (valid percent) of adult females and males meeting the PA guideline were acquired from the Dwyer-Lindgren et al. online supplemental material [28]. Specifically, the dependent variables were the 2011 BRFSS estimates for females and males, predicted using a small area estimation model that included race/ethnicity, education, poverty, unemployment, air pollution, rural-urban status covariates, and a geospatial term. These data are the most recent county-level estimates of meeting PA guidelines available for the entire U.S. For further information on the small area estimation methodology see Dwyer-Lindgren et al. [28] and Srebotnjak, Mokdad, and Murray [29].

#### 2.3.2. Environment Factors

The independent/predictor variables were categorized into natural, social, recreation, and transportation environments in alignment with Sallis and colleagues’ [6] framework. Some variables are measures of the built environment (e.g., park proximity), some are behavioral proxies for environmental measures (e.g., commuting by bike), and some are objective measures that may impact the perceived barriers to engaging in PA (e.g., crime, heat).

### 2.4. Statistical Analysis

All analyses were performed in SAS 9.4^©^ (SAS Institute, Cary, NC, USA) [30] with missing observations handled using pairwise techniques (missing patterns 1%). Univariate and bivariate analyses were used to examine measures of central tendency and correlational associations between study variables. Analyses were conducted in two steps: (1) developing a parsimonious model, and (2) testing for associations of model factors and the dependent variable.

Factor Analyses (FA) were used to establish model parsimony, variable redundancy, and obtain weighted factor scores prior to regression analyses on the dependent variables. First, all variables were standardized into z-scores, reviewed for non-normality, and log-transformed (when appropriate) to ensure statistical assumptions for inferential analyses were met. FA criteria used squared multiple correlations (SMC) as priors for communality estimation, an orthogonal (varimax) rotation, and loadings ≥40 considered “large” for factor extraction. Lastly, estimated factor scores were assigned to each observation using the linear composite from the observed variables [31].

Multiple Linear Regression (MLR) analyses were used to identify associations between principal components (factor scores) and county-level prevalence of males and females meeting PA guidelines. MLR models were stratified by quintile and accounted for clustering effects by state. Standardized (β) beta coefficients and standard errors (SE) represent model association. Lastly, model assumptions as outlined by [32] were performed prior to MLR with alphas of ≤0.05 considered statistically significant. 

## 3. Results

### 3.1. Descriptive County Characteristics

Univariate analyses are presented in Table 2. Across all counties, the average prevalence of meeting PA guidelines was 52.8% among males and 49.0% among females, ranging from 56.2% for male residents of urban counties to 46.8% among female residents of rural counties. Bivariate correlations for male and female PA for the indicators and each of the environment factors (natural, social, recreation, and transportation) are presented in Appendix A. Those results demonstrate that male and female PA correlate similarly with all 18 of the indicators in each of the four environment factors across the rural-urban continuum. 

### 3.2. Factor Loadings and Model Parsimony 

Factor loadings are reported in Table 3. FAs extracted a single factor for each environmental construct. Variable redundancy and loading criteria resulted in a reduction from 18 to 13 variables. The FA solution suggests factors comprised of natural environment (Air Temperature, Sun, and Heat Index), social environment (Single Parent Households, Violent Crime, and Vacant Housing), recreation environment (Proximity to a Park, Access to Exercise Opportunities, and Proximity to an Elementary School), and transportation environment (Walk to Work, Works in County, Works in Place, and Long Commute). Of note are the low factor loading (0.40) of Access to Exercise Opportunities and Bike to Work in the *Somewhat Rural*, *Rural*, and *Exclusively Rural* counties. For this reason, and to further support model parsimony, these variables were not included in the final model.

### 3.3. Associations between PA and Factor Scores by Sex

Results from the MLR analyses by sex are presented in Table 4. For males, the analyses reveal associations between natural environment (β = −32, *p* 0.01), social environment (β = −11, *p* 0.05), and recreation environment (β = 0.34, *p* 0.01). Female PA was associated with the natural environment (β = −31, *p* 0.01), social environment (β = −24, *p* 0.01), and recreation environment (β = 0.33, *p* 0.01). The transportation environment was not associated with PA for either sex (β = 0.01, *p* 0.05). 

### 3.4. Associations between PA, and Factor Scores by Rurality and Sex

Results from the MLR analyses by rurality and sex are also presented in Table 4. Overall, the pattern of associations across the rural-urban continuum is similar between the sexes. Negative associations are seen for natural environment factors in all levels of rurality (β −36 to −46; all *p* 0.01) except the *Urban* counties. The recreation environment was associated with meeting PA guidelines in the *Urban* category of counties and the two most rural categories, with a greater difference in β between rural and urban counties among males than females. The negative association of the social environment and PA was significant among men except in the two most rural categories of counties; among females, it was also significant in the *Exclusively Rural* counties (β = −20, *p* 0.05). The transportation environment was associated with PA only in females in *Somewhat Urban* counties (β = 0.12, *p* 0.05).

## 4. Discussion

Findings from this study address a critical research gap highlighted in reviews of the literature [9,10] by identifying variations in the strength of environmental factors associated with meeting PA guidelines across the rural-urban continuum, using objective measures for—and an inclusive definition of—“environment” elucidated by Sallis and colleagues in their seminal work [6]. The natural environment factor, including air temperature, heat index, and precipitation, emerged as a barrier to PA, especially in rural areas. While long recognized by public health professionals for its role in communicable disease [33], this study highlights a mechanism by which climate may limit PA and lead to chronic disease, especially in rural areas of the southern and southeastern U.S. that have a lower prevalence of PA than other regions [3] and increasing experience of extreme climate events [34]. These events, such as wildfires, tornadoes, and flooding, may be particularly impactful in rural areas. For example, poor air quality from regional wildfires was found to have a greater negative impact on a walking intervention among rural participants than urban ones [35]. In addition to such acute effects, extreme weather events may cause lasting damage or destruction to the limited number of outdoor recreation facilities available in rural areas, where financial resources for reconstruction may be limited.

The association of the recreation environment with meeting PA guidelines was strongest in the *Rural* and *Exclusively Rural* counties. Within this factor, access to exercise opportunities (including private recreation facilities) did not load significantly in the two most rural categories of counties, suggesting the other variables—proximity to parks and elementary schools—are particularly important as rurality increases, in concordance with existing literature [8,9,10]. This is important because access to and funding of these facilities is a function of local decision making (e.g., shared use of school facilities, parks and recreation department budgets) [36]. In the *Urban* counties, access to places to exercise was of particular importance, demonstrating the primacy that private and/or indoor facilities may have in high-density urban areas. In combination with the natural environment results, these findings highlight the importance of creating opportunities for low-to-no-cost indoor activity, providing shaded outdoor places, and/or utilizing natural water assets (e.g., lakes, rivers, swimming holes) or swimming pools for rural PA, in concurrence with existing research [9].

The social environment was negatively associated with meeting PA guidelines, with greater regression coefficients in urban counties than rural ones, and a coefficient for females twice that of males across all levels of rurality. This factor was composed of variables serving as proxy measures for safety and social barriers to PA, with the prevalence of violent crime and single-parent households loading most significantly on the factor across all levels of rurality and for both sexes. Our findings, using objective measures, add to the preponderance of evidence observing that safety is associated with PA in rural settings [8]. The literature regarding the differential impact of social role constraints such as single parenting on female’s PA, especially rural females, is less clear. For example, Dlugonski and colleagues [37] identified chronic stress from being a caregiver and from meeting children’s basic needs as barriers to PA in focus groups of low-income Black single mothers of unknown rurality. Additionally, Eyler and colleagues [38] found that having fewer social role strains (e.g., child/eldercare, household tasks, work) was associated with meeting PA guidelines among rural white females (15.6% unpartnered, parenting status unknown) but not rural African American females (40.5% unpartnered, parenting status unknown) [38,39,40]. Our results add to this evolving aspect of the literature, suggesting that sociocultural PA barriers such as crime and single parenting (1) are more impactful on a female’s PA than on a male’s, and (2) may reduce in influence on PA as rurality increases until reaching the *Most Rural* counties; additional understanding of these relationships is needed.

The transportation environment factor had a nearly zero correlation with the adult prevalence of meeting PA guidelines, except for a negative association with PA among females in *Somewhat Urban* counties. This suggests that commuting behaviors and living near work were not influential in meeting PA guidelines when considering a comprehensive model of multiple environmental factors. This is consistent with other data from rural areas where transportation PA is less likely [9,10], but is somewhat unexpected for urban areas where transportation PA is more likely [41]. However, this lack of association should be considered with caution because the dependent variable in our analyses was *leisure time* PA as measured by the BRFSS and did not include occupational, household, or transportation PA. Recent analyses of 2011–2016 National Health and Nutrition Examination Survey data by Whitfield and colleagues [41], using domain-specific PA for rural and urban adults, demonstrated that a lower prevalence of meeting PA guidelines among rural adults is attenuated when considering occupational and household PA, in addition to leisure and transportation PA that are more common in urban adults.

There are multiple limitations to the current study worth noting. First, the PA data utilized are 10 years old, based on a small area estimation, and include only aerobic and leisure-time PA. Though PA guidelines have since been revised to incorporate muscle-strengthening PA [42], and research now highlights the importance of household [43] and occupational PA [41,43] for rural adults, these are the most recent data for meeting PA guidelines for every county in the U.S. Second, we chose to exclude frontier counties from the analyses. We excluded frontier counties primarily because the environmental measures we used in our analyses were largely missing from these counties or had extremely wide variation because of small sampling in each county. We encourage in-depth study of PA in those 443 frontier counties which have such unique characteristics. Third, because factors known to be associated with PA were included in the Dwyer-Lindgren small area estimation of PA prevalence (e.g., individual-level race/ethnicity, age group, county-level race, poverty, education), we could not assess their influence in our analysis [28]. PA prevalence data at the county level for all counties in the US continues to be a necessity and should be a priority in future national surveillance (e.g., BRFSS) and localized data reporting (e.g., CDC PLACES).

The findings of this study highlight multiple future needs, to (1) improve the granularity of PA surveillance in rural areas of the U.S., (2) identify the facilitators of and barriers to adults’ PA in rural areas of the US, and (3) utilize evidence-based strategies from rural research to address the PA disparities evident in rural areas of the U.S. Building this rural PA evidence will require a more detailed understanding of the variation of environmental factors associated with adults’ PA within rural communities. Positive Deviance methods have been used in under-resourced communities to identify promising practices in places that have better-than-expected outcomes (“Positive Outliers”) than peers so that the practices can be disseminated to peer communities [44,45]. Used most often to identify promising nutrition practices in developing countries, it has only recently been applied to identify promising PA practices among African American females [46]. The first step in this approach is identifying the places that are outperforming peers on the measures of importance. The data used for the analyses presented herein provide a promising first step in that process. Additional studies of PA in rural U.S. areas focusing on specific age groups are also warranted because rural areas of the U.S. tend to have a larger proportion of older adults than urban and suburban areas [47]. Further, though our study focused exclusively on the U.S., comparative international rural PA research is also warranted. Like the U.S.-based built environment PA literature, comparative international studies, such as those conducted by the International Physical Activity and Environment Network (IPEN), have initially focused on urban areas where large sample sizes and datasets are more readily accessible [48,49,50,51].

## 5. Conclusions

This comprehensive, national-level comparison of environmental factors associated with adults’ PA in rural and urban areas in the U.S. addresses a gap previously identified in the literature [9,10]. Despite the limitations noted, the findings highlight the variation of natural, recreation, and social environment factors associated with PA across the rural-urban continuum by sex. Replication of our analyses with more recent data is needed to determine whether these associations have persisted or changed over time. 

## Figures and Tables

**Table 1 ijerph-18-07688-t001:** List of Data Sources and Variables by Environment Type.

Variable	Source	Definition (Scale)
Dependent Variable: Prevalence of meeting PA Guidelines		
Female PA	2011 BRFSS [27]	150 min per week of MPA, 75 min per week of VPA, or combination of MVPA (1 min VPA = 2 min MPA), (% of females) [28]
Male PA	2011 BRFSS [27]	150 min per week of MPA, 75 min per week of VPA, or combination of MVPA (1 min VPA = 2 min MPA), (% of males) [28]
Natural Environment		
Air Temperature	2011 WONDER [26]	Average Daily Max Air Temperature (F°)
Heat Index	2011 WONDER [26]	Average Daily Max Heat Index (F°)
Sunlight	2011 WONDER [26]	Average Daily Sunlight (KJ/m^2^)
Precipitation	2011 WONDER [26]	Average Daily Precipitation (mm)
Water area	2010 Census [21]	Total Area of County that is Water (%)
Social Environment		
Alcohol Vehicle Deaths	2012, 5 year FARS (from 2014 CHR) [23]	Motor vehicle crash deaths with alcohol involvement (%)
Violent Crime	2011, 3 year UCR (from 2014 CHR) [23]	Number of violent crimes per 100,000 population
Single Parent Households	2012, 5 year ACS (from 2014 CHR) [23]	Children 18 years of age living in family households that are headed by a single parent (%)
Vacant Housing	2011, 5 year ACS (from 2011 EPH) [25]	Housing units that are vacant (%)
Recreation Environment		
Access to Exercise Opportunities	2014 CHR (using 2010 and 2012 data) [23]	Population within ½ mile of a park or 1 mile (urban) or 3 miles (rural) of a recreation facilities (%)
Highway proximity	2010 EPH [25]	Population living within 150 m of a highway (%)
Park proximity	2010 EPH [25]	Population living within Half-Mile of Park (%)
School proximity	2010 EPH [25]	Population aged 5 to 9 years living within a half mile of a public elementary school (%)
Transportation Environment		
Bike to Work	2011, 5 year ACS (from 2011 EPH) [25]	Workers over 16 years that bike to work (%)
Walk to Work	2011, 5 year ACS (from 2011 EPH) [25]	Workers over 16 years that walk to work (%)
Works in County	2012, 5 year ACS (from 2014 CHR) [23]	Works in County of Residence (%)
Works in Place	2012, 5 year ACS (from 2014 CHR) [23]	Works in Place of Residence (%)
Long Commute	2012, 5 year ACS (from 2014 CHR) [23]	Drive Alone-Long Commute to Work (%)
Cluster and Stratification		
Rural Percent	2010 Census [21]	Population that lives in a rural area (%)
Rural Quintile (group)		
State (name)	2010 Census [21]	

Notes: F°, Fahrenheit; KJ/m^2^, kilojoules per square meter; mm, millimeters; MPA, moderate physical activity; VPA, vigorous physical activity; MVPA, moderate-to-vigorous physical activity; ACS, American Community Survey, (https://www.census.gov/data.html, accessed on 23 February 2020); BRFSS, Behavioral Risk Factor Surveillance System, (https://www.cdc.gov/brfss/, accessed on 23 February 2020); CDC, Centers for Disease Control and Prevention; CHR, County Health Rankings, (https://www.countyhealthrankings.org/, accessed on 23 February 2020); WONDER, Wide-ranging Online Data for Epidemiologic Research, (https://wonder.cdc.gov/nasa-nldas.html, accessed on 28 July 2020); EPH, National Environmental Public Health Tracking Network, (https://ephtracking.cdc.gov/, accessed on 28 July 2020); UCR, The Uniform Crime Reporting Program, (https://www.fbi.gov/services/cjis/ucr, accessed on 23 February 2020).

**Table 2 ijerph-18-07688-t002:** Measures of Central Tendency for 2011 Male and Female County-level Prevalence of Meeting Physical Activity Guidelines.

		Male PA						Female PA					
% Rural	Group	*n*	%	M	SD	Min	Max	*n*	%	M	SD	Min	Max
0–25	Urban	577	21.4	56.2	15.9	36.8	72.7	577	21.4	52.6	21.6	33.1	74.2
25–48	Somewhat Urban	577	21.4	54.1	13.7	38.4	73.2	577	21.4	50.2	19.7	30.7	74.7
48–67	Somewhat Rural	577	21.4	52.2	16.3	37.5	72.2	577	21.4	48.0	21.9	29.1	68.8
67–99	Rural	577	21.4	50.6	20.2	33.7	71.4	577	21.4	46.8	25.5	29.0	67.8
100	Exclusively Rural	389	14.4	50.1	21.5	33.1	70.4	389	14.4	46.9	26.0	31.3	71.6
	All	2697	100.0	52.8	19.2	33.1	73.2	2697	100.0	49.0	24.4	29.0	74.7

Notes: The data source was Dwyer-Lindgren and colleagues’ [28] county-level small area estimation using 2011 Behavioral Risk Factor Surveillance System data. The US PA guidelines at the time were for adults to achieve 150 min of moderate intensity PA per week, the equivalent in vigorous PA, or combination of moderate and vigorous PA where one minute of vigorous PA is equivalent to two minutes of moderate PA.

**Table 3 ijerph-18-07688-t003:** Exploratory Factor Analysis Factor Loadings.

	All Groups	Urban	Somewhat Urban	Somewhat Rural	Rural	Exclusively Rural
		(0–25% Rural)	(25–48% Rural)	(48–67% Rural)	(67–99% Rural)	(100% Rural)
Natural Environment (*n*)	(2681)	(567)	(573)	(577)	(577)	(387)
Air	**0.98**	**0.96**	**0.97**	**0.98**	**0.99**	**0.99**
Sun	**0.87**	**0.90**	**0.84**	**0.89**	**0.90**	**0.85**
Heat	**0.66**	**0.52**	**0.64**	**0.72**	**0.76**	**0.70**
Water	−0.08	0.00	−0.07	−0.19	−0.19	−0.08
Precipitation	−0.21	−0.42	−0.22	−0.24	−0.13	0.12
Social Environment (*n*)	(2577)	(571)	(561)	(554)	(554)	(337)
Single Parent Households	**0.67**	**0.80**	**0.68**	**0.73**	**0.61**	**0.47**
Violent Crime	**0.59**	**0.75**	**0.67**	**0.68**	**0.57**	**0.53**
Vacant Housing	0.24	**0.49**	0.36	**0.48**	0.34	**0.42**
Alcohol Vehicle Deaths	0.07	0.00	0.05	0.14	0.05	0.16
Recreation Environment (*n*)	(2686)	(576)	(576)	(576)	(577)	(381)
Live within Half-Mile of Park	**0.76**	**0.75**	**0.67**	**0.61**	**0.65**	**0.58**
Access to Exercise Opportunities	**0.59**	**0.58**	**0.41**	0.29	0.25	0.10
Elementary Half-Mile	**0.55**	**0.57**	**0.54**	**0.51**	**0.57**	**0.56**
Live within 150 Miles of Highway	**0.46**	0.32	0.12	0.08	0.01	0.14
Transportation Environment (*n*)	(2697)	(577)	(577)	(577)	(577)	(389)
Works in County	**0.82**	**0.71**	**0.82**	**0.82**	**0.82**	**0.79**
Works in Place	**0.75**	**0.70**	**0.74**	**0.69**	**0.65**	**0.68**
Walk to Work	**0.50**	**0.53**	**0.60**	**0.47**	**0.53**	**0.56**
Bike to Work	**0.49**	**0.47**	**0.52**	0.37	0.36	0.35
Long Commute	**−0.81**	**−0.71**	**−0.82**	**−0.80**	**−0.76**	**−0.82**

Notes: Factor scores ≥ 0.4 are bolded for emphasis.

**Table 4 ijerph-18-07688-t004:** Standardized Regression Coefficients and Standard Errors for Multiple Linear Regression of 2011 Male and Female Prevalence of Meeting Physical Activity Guidelines by Environmental Factor Scores.

	Male Physical Activity											
	All Groups (*n* = 2560)		Urban (*n* = 564)		Somewhat Urban (*n* = 557)		Somewhat Rural (*n* = 553)		Rural (*n* = 554)		Exclusively Rural (*n* = 332)	
Factor	β	se	β	se	β	se	β	se	β	se	β	se
Natural Environment	−0.32 ^a^	0.07	−0.17	0.10	−0.41 ^a^	0.08	−0.46 ^a^	0.09	−0.46 ^a^	0.11	−0.43 ^a^	0.14
Social Environment	−0.11 ^b^	0.07	−0.32 ^a^	0.08	−0.19 ^a^	0.08	−0.16 ^b^	0.08	0.01	0.11	−0.07	0.10
Recreation Environment	0.34 ^a^	0.06	0.12 ^b^	0.14	0.01	0.08	0.07	0.10	0.24 ^a^	0.09	0.28 ^b^	0.14
Transportation Environment	−0.01	0.03	−0.07	0.06	0.05	0.05	−0.05	0.06	0.01	0.05	−0.06	0.09
	**Female Physical Activity**											
	All Groups (*n* = 2560)		Urban (*n* = 564)		Somewhat Urban (*n* = 557)		Somewhat Rural (*n* = 553)		Rural (*n* = 554)		Exclusively Rural (*n* = 332)	
Factor	β	se	β	se	β	se	β	se	β	se	β	se
Natural Environment	−0.31 ^a^	0.08	−0.16	0.13	−0.36 ^a^	0.09	−0.42 ^a^	0.11	−0.46 ^a^	0.13	−0.41 ^a^	0.13
Social Environment	−0.24 ^a^	0.06	−0.43 ^a^	0.08	−0.31 ^a^	0.08	−0.30 ^a^	0.08	−0.09	0.12	−0.20 ^b^	0.11
Recreation Environment	0.33 ^a^	0.07	0.20 ^b^	0.15	0.07	0.10	0.11	0.09	0.27 ^a^	0.09	0.27 ^b^	0.14
Transportation Environment	0.01	0.03	−0.03	0.06	0.12 ^b^	0.05	−0.01	0.05	−0.02	0.06	−0.02	0.09

Notes: ^a^: *p* ≤ 0.01, ^b^: *p* ≤ 0.05. Regression models were adjusted for counties clustered by state.

## Data Availability

No new data were created or analyzed in this study. Data sharing is not applicable to this article.

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
