# Peer review of "Environmental Factors Associated with Physical Activity in Rural U.S. Counties"

_ijerph, 2021, doi:10.3390/ijerph18147688_

Round 1
Reviewer 1 Report
This is a very interesting and well written manuscript. It is well structures and methodologically robust. I only have three remarks:
- The references used in the introduction and the discussion could be more recent in order to justify the study. I would suggest the authors to take a look at this article: Santinha, G, Wolf, J & Costa, C (2020) Aging and the built environment: is mobility constrained for institutionalized older adults?, Journal of Urbanism: International Research on Placemaking and Urban Sustainability, 13:4, 431-447.
- The limitations referred to by the authors are indeed important and, because of that, it diminishes the added value of the results.
- The findings highlight the variation of multiple environmental factors associated with PA across the rural-urban continuum by sex. However, nothing is said related to age. This is an important lack of the study. If such data is available, the authors should include it in the analysis.
Author Response
Reviewer #1:
This is a very interesting and well written manuscript. It is well structures and methodologically robust. I only have three remarks:
The references used in the introduction and the discussion could be more recent in order to justify the study. I would suggest the authors to take a look at this article: Santinha, G, Wolf, J & Costa, C (2020) Aging and the built environment: is mobility constrained for institutionalized older adults?, Journal of Urbanism: International Research on Placemaking and Urban Sustainability, 13:4, 431-447.
- Response: We appreciate this suggestion. We agree that our key citations justifying the study are 5-10 years old (Frost, Umstattd Meyer, Hansen) and hope our more recent citations of work by Whitfield and Carlson from the last few years also help the reader understand the persistence of the rural-urban differences in physical activity in the US. We also acknowledge that this study is focused exclusively on the US, with reasons and edits to the text described in the response to subsequent comments by each of the reviewers.
The limitations referred to by the authors are indeed important and, because of that, it diminishes the added value of the results.
- Response: We agree and hope we adequately acknowledged those limitations.
The findings highlight the variation of multiple environmental factors associated with PA across the rural-urban continuum by sex. However, nothing is said related to age. This is an important lack of the study. If such data is available, the authors should include it in the analysis.
Response: We appreciate the positive feedback and suggestions from Reviewer #1. Reviewer #2 also noted (1) the lack of acknowledgement of older adults and their needs in the built environment, and (2) a lack of international focus (as evidenced by the citation this reviewer provided). These are excellent points. Our focus was not on age-specific physical activity because we did not have access to age-specific PA data. Nor was it on a comparison between countries. However, because rural areas of the US generally have a higher prevalence of older adults than urban areas and because of the differences likely to be revealed by comparing rural US adults & rural adults in other countries, we have made edits to the final paragraph of the discussion section to enhance our suggestions about potential future research.
Reviewer 2 Report
Dear authors,
First of all, I would like to congratulate you on this work, as a
result of the effort, time and dedication that this entails.
Second, and after reading the document, I am going to give you some
recommendations regarding the introduction, methodology and conclusions. Regarding the introduction, I suggest that you provide more
conceptualization or talk more about the studies on the differences that
can influence the practice of physical activity at a rural or urban level.
I notice a lack of depth in your introduction regarding the differences,
alluding to previous studies. I recommend you better conceptualize your
introduction, incorporating studies with dedicated authors.
I highlight studies with authors that deal with their theme regarding
active aging and physical activity on the line
https://pubmed.ncbi.nlm.nih.gov/32085450/
and that investigates on active aging, and on physical activity,
or this other https://pubmed.ncbi.nlm.nih.gov/34203410/
I also recommend that you specify at the end of the introduction the
importance of investigating the differences between rural and urban areas.
Conceptualize them, define them for the reader. I suggest you delve into
the differences between rural and urban activity, what benefits or harms
each of them has.
Regarding the methodology: In the evaluation instruments, specify more
each one of them, referring to an example of the item that evaluates,
scales, dimensions, scores.
Regarding discussion / conclusion, I recommend incorporating the following
points: What are the implications of this work for the scientific community? What implications does this work have for society on a practical level? What future prospects or future lines can be drawn from here. Even so, for my part they are suggestions to improve your manuscript,
congratulate you again on your work and time.
All the best,
Author Response
Reviewer #2
First of all, I would like to congratulate you on this work, as a result of the effort, time and dedication that this entails. Second, and after reading the document, I am going to give you some recommendations regarding the introduction, methodology and conclusions. Regarding the introduction, I suggest that you provide more conceptualization or talk more about the studies on the differences that can influence the practice of physical activity at a rural or urban level. I notice a lack of depth in your introduction regarding the differences, alluding to previous studies. I recommend you better conceptualize your introduction, incorporating studies with dedicated authors. I highlight studies with authors that deal with their theme regarding active aging and physical activity on the line https://pubmed.ncbi.nlm.nih.gov/32085450/ and that investigates on active aging, and on physical activity, or this other https://pubmed.ncbi.nlm.nih.gov/34203410/
- Response: We appreciate the positive feedback and suggestions. Reviewer #1 also noted (1) the lack of acknowledgement of older adults and their needs in the built environment, and (2) a lack of international focus (as evidenced by the citations this reviewer provided). These are excellent points. Our focus was not on age-specific physical activity because we did not have access to age-specific PA data. Nor was it on a comparison between countries. However, because rural areas of the US generally have a higher prevalence of older adults than urban areas and because of the differences likely to be revealed by comparing rural US adults & rural adults in other countries, we have made edits to the final paragraph of the discussion section to enhance our suggestions about potential future research.
I also recommend that you specify at the end of the introduction the importance of investigating the differences between rural and urban areas. Conceptualize them, define them for the reader. I suggest you delve into the differences between rural and urban activity, what benefits or harms each of them has.
- Response: We appreciate this comment and the previous one that is related (“I suggest that you provide more conceptualization or talk more about the studies on the differences that can influence the practice of physical activity at a rural or urban level”). We hope that the introduction as currently structured meets the reviewer’s needs. First, we chose to keep the introduction at a fairly high level because, as noted in the second paragraph of the introduction, the uniqueness of rural areas and the importance of investigating the differences between rural and urban areas has been discussed in multiple systematic reviews and “call to action” papers that highlight the research gap that our current study is attempting to fill. We did, however, provide a few examples in paragraph 3 of the introduction about “unique challenges” of physical, natural, and sociocultural factors on PA in rural areas. Second, we chose to begin the introduction with the differences between rural and urban physical activity from recent epidemiological studies and a statement in the first sentence that mentions that there are health disparities evident in rural US and, rather than restate all of them, we chose to dive right into our topic of physical activity because of the topic of the special issue.
Regarding the methodology: In the evaluation instruments, specify more each one of them, referring to an example of the item that evaluates, scales, dimensions, scores.
- Response: We appreciate this comment but hope the current description with external citation for full methods will satisfy the reviewer. We reviewed papers that used multiple national datasets (eg, reference #18 by Kaczynski and colleagues) for a similar purpose that structure their manuscript similarly. Our choice was made to be as parsimonious as possible while also providing the reader the opportunity to easily find full details of the datasets elsewhere.
Regarding discussion / conclusion, I recommend incorporating the following points: What are the implications of this work for the scientific community? What implications does this work have for society on a practical level? What future prospects or future lines can be drawn from here. Even so, for my part they are suggestions to improve your manuscript, congratulate you again on your work and time.
Response: Thank you so much for the recommendations. We have revised the final paragraph of the Discussion section to enhance our recommendations for future research, including age-specific research and international comparative studies of rural places. Such studies would be fascinating! Urban-centric studies are underway targeting adults and adolescents as part of the International Physical Activity and Environment Network (IPEN), but are needed for rural areas, as well.
Reviewer 3 Report
Review of the article Environmental Factors Associated with Physical Activity in Rural US Counties
This article provides an analysis of the environmental factors associated with PA in adults in the US. It is a very interesting report showing the differences between urban and rural areas.
Introduction
This section lacks information on analyzes and research from other countries (outside the US), which may provide a broader view of the problem. Materials and Methods In this section, the authors describe each part of the procedures and source data well. But if the title of the article and the aim of the work mention physical activity, at least basic methodological information should be provided, how was the PA index analyzed? (at the moment there is only a link in section 2.2. Where to find more information ...).
2.3.1. Prevalence of Physical Activity
The situation is similar in this section where the authors write about the guidelines from Dwyer-Lindgren et al. - but it is not known what! In a few sentences, I suggests explaining which compliance guidelines are meant and what methods of PA analysis are meant. These guidelines may be recognizable in research in the USA, but less so in other countries.
Results
In table 2, he suggests to fill in that it is about meeting the criterion of e.g. 150 minutes of PA a week (according to The NHANES).
Very good analysis in subsections 3.2, 3.3 and 3.4
References
In this part of the work it is also noticeable that it is narrowed down to one area (USA).
In subsequent analyzes, he suggests analyzing qualitative tests (pedometers, accelerometers ...), for example in the research: Herbert, J .; Matłosz, P .; Lenik, J .; Glider, A .; Aries, J .; Przednowek, K .; Wyszyńska, J. Objectively Assessed Physical Activity of Preschool-Aged Children from Urban Areas. Int. J. Environ. Res. Public Health 2020, 17, 1375 Quantitative research, among others using The NHANES are very good, but it should be remembered that it is the respondents who define their physical activity by often overestimating it (overestimation occurs).
Author Response
Reviewer #3
This article provides an analysis of the environmental factors associated with PA in adults in the US. It is a very interesting report showing the differences between urban and rural areas.
Introduction
This section lacks information on analyzes and research from other countries (outside the US), which may provide a broader view of the problem.
- Response: We appreciate this feedback. It was evident in all three reviewers’ comments that the paper lacked an international focus. This is an excellent point. Our purpose was to address an important research gap in the US by comparing between rural and urban areas within the US. We hope this leads to subsequent research comparing rural areas <between countries> and have made note of that in the final paragraph of the discussion section to enhance our suggestions about potential future research. There is precedent for international comparative studies: urban-centric studies are underway targeting adults and adolescents as part of the International Physical Activity and Environment Network (IPEN). Replicating such work in rural areas would answer fascinating additional questions about the varying influence of physical, social, cultural, natural, built, and recreational environment factors affecting rural residents’ PA across countries.
Materials and Methods
In this section, the authors describe each part of the procedures and source data well. But if the title of the article and the aim of the work mention physical activity, at least basic methodological information should be provided, how was the PA index analyzed? (at the moment there is only a link in section 2.2. Where to find more information ...).
- Response: Please see subsequent comment regarding revisions to section 2.3.1 to provide details of the relevant items in the data source for the PA variable
2.3.1. Prevalence of Physical Activity
The situation is similar in this section where the authors write about the guidelines from Dwyer-Lindgren et al. - but it is not known what! In a few sentences, I suggests explaining which compliance guidelines are meant and what methods of PA analysis are meant. These guidelines may be recognizable in research in the USA, but less so in other countries.
- Response: Thank you for this comment. We have revised section 2.3.1 to insert the details about the items used in the BRFSS to assess individual respondents’ PA minutes, whether they met PA guidelines or not, and what the PA guidelines were at the time. We also revised section 2.2.2 to clarify that the BRFSS is telephone-based and is thus self-report data.
Results
In table 2, he suggests to fill in that it is about meeting the criterion of e.g. 150 minutes of PA a week (according to The NHANES).
- Response: Thank you for this comment. A note was added to table 2 to better describe the data source and PA guidelines.
Very good analysis in subsections 3.2, 3.3 and 3.4
- Response: Thank you!
References
In this part of the work it is also noticeable that it is narrowed down to one area (USA).
In subsequent analyzes, he suggests analyzing qualitative tests (pedometers, accelerometers ...), for example in the research: Herbert, J .; Matłosz, P .; Lenik, J .; Glider, A .; Aries, J .; Przednowek, K .; Wyszyńska, J. Objectively Assessed Physical Activity of Preschool-Aged Children from Urban Areas. Int. J. Environ. Res. Public Health 2020, 17, 1375 Quantitative research, among others using The NHANES are very good, but it should be remembered that it is the respondents who define their physical activity by often overestimating it (overestimation occurs).
Response: We appreciate this comment. We made note in section 2.2.2 that the data source for the PA outcome in our analyses was self-report via the BRFSS, and hope that our future research suggestions in the last paragraph of the Discussion section help clarify that we are making suggestions about potential research methods for better understanding what works in creating physically active rural areas (qualitative, likely case studies) rather than making suggestions for improving physical activity surveillance data collection to reduce the overestimation of PA that is typical of self-report.